# Mitochondrial Dysfunction in the Aging Retina

**DOI:** 10.3390/biology8020031

**Published:** 2019-05-11

**Authors:** Janis T. Eells

**Affiliations:** Department of Biomedical Sciences, University of Wisconsin-Milwaukee, Milwaukee, WI 53211, USA; jeells@uwm.edu; Tel.: +1-608-215-5405

**Keywords:** aging, mitochondria, retina, optic nerve, diabetic retinopathy, age-related macular degeneration, glaucoma

## Abstract

Mitochondria are central in retinal cell function and survival and they perform functions that are critical to cell function. Retinal neurons have high energy requirements, since large amounts of ATP are needed to generate membrane potentials and power membrane pumps. Mitochondria over the course of aging undergo a number of changes. Aged mitochondria exhibit decreased rates of oxidative phosphorylation, increased reactive oxygen species (ROS) generation and increased numbers of mtDNA mutations. Mitochondria in the neural retina and the retinal pigment epithelium are particularly susceptible to oxidative damage with aging. Many age-related retinal diseases, including glaucoma and age-related macular degeneration, have been associated with mitochondrial dysfunction. Therefore, mitochondria are a promising therapeutic target for the treatment of retinal disease.

## 1. Retinal Anatomy and Physiology

The fundamental organization of the retina is conserved across vertebrates. The retina contains five major neuronal cell classes (photoreceptors, bipolar cells, amacrine cells, horizontal cells, and retinal ganglion cells) with Müller glial cells and retinal pigment epithelial cells providing metabolic and homeostatic support [1] (Figure 1). The interaction of light with light-sensitive pigments in the outer segments of rod and cone photoreceptors initiates signaling mechanisms that convert light energy to neural activity. Photoreceptors synapse with bipolar cells. Bipolar cells synapse with the retinal ganglion cells that transmit the visual signal along their axons (optic nerve) and, ultimately, to the brain. The retinal pigment epithelium envelopes the photoreceptors, which facilitates the diffusion of nutrients, key metabolites, and oxygen from the choroidal vessels. Horizontal cells, amacrine cells, and Muller cells mediate and support the synapses between the retinal neurons. Unique to the Muller cell, its processes span the inner retina and most of the outer retina, assisting in the maintenance of multiple synapses throughout the retina. Pericytes surround the vascular endothelial cells in all retinal layers, forming the blood-retinal barrier.

The retina is the highest oxygen-consuming organ in the human body [2,3,4,5,6]. The inner retinal neurons have the highest metabolic rate of all central nervous tissue, and the oxygen consumption rate of the photoreceptors is several times higher again. The inner segments of the photoreceptor cells are rich in mitochondria providing the ATP that are required by the ionic pumps that drive the ‘dark current’. In addition, photoreceptors also metabolize glucose through aerobic glycolysis [2,3,4,5,6].

The key role of aerobic glycolysis in the vertebrate retina has been long recognized and well established [2,5,6]. The majority of aerobic glycolysis occurs in the photoreceptors. Although aerobic glycolysis dominates energy production in the photoreceptors, the retina requires both glycolysis and oxidative phosphorylation to initiate vision [4]. This has been demonstrated in studies showing that the responses of neurons downstream of photoreceptors are abolished by inhibiting glycolysis or by removing O_2_ [2]. Recent studies have proposed a model in which glucose from the choroidal blood passes through the retinal pigment epithelium to the retina, where photoreceptors convert it to lactate [6]. Photoreceptors then export the lactate as fuel for the retinal pigment epithelium and for neighboring Müller glial cells. Aerobic glycolysis not only enables enhanced anabolic metabolism in photoreceptors, but also maximizes their function and adaptability to nutrient stress conditions.

Photoreceptors consume more oxygen per gram of tissue weight than any cell in the body [2,5]. This intense degree of oxidative phosphorylation in their inner segments, coupled with high concentrations of polyunsaturated fatty acids in their outer segments, renders the retina susceptible to oxidative stress and lipid peroxidation [2,3,4]. The retina is exposed to visible light and it contains abundant photosensitizers [1]. Normally, endogenous antioxidants and repair systems minimize oxidative damage. However, with aging and retinal disease, there is an increase in mitochondrial dysfunction and oxidative damage and corresponding decrements in antioxidants and repair systems, resulting in retinal dysfunction and retinal cell loss, which leads to visual impairment [7,8,9,10,11,12,13].

## 2. Mitochondrial Dysfunction with Aging and Disease

### 2.1. Age-Related Mitochondrial Changes

Mitochondria are central to retinal cell function and survival and they perform a variety of functions critical to cell metabolism, including oxidative phosphorylation, beta-oxidation of fatty acids, calcium homeostasis, and the regulation of neuronal excitability [7,8,9,10,11,12,13]. Retinal neurons have high energy requirements and large amounts of ATP are needed to generate membrane potentials and power membrane pumps. It has been estimated that 90% of mitochondria-generated ATP is used to maintain transmembrane ion gradients [10].

Mitochondria in the aging retina show a decline in the function of the electron transport chain and a significant increase in the generation of reactive oxygen species (ROS) (Figure 2). This increased ROS production leads to increased oxidative damage to mitochondrial DNA, lipids, and proteins [5,6,14,15,16]. With aging, there is an accumulation of mtDNA mutations and a disruption of mitochondrial structure. Mitochondrial genomic instability has been postulated to play an important role in age-related retinal pathophysiology [16]. However, murine studies have shown that the majority of mtDNA mutations in aged cells appear to be caused by replication errors early in life, rather than by oxidative damage [15]. Mitochondrial DNA repair pathways are limited and they begin to fail with aging, leading to additional mtDNA damage and contributing to the development of retinal degenerative diseases [8,9].

### 2.2. Age-Related Macular Degeneration

Age-related macular degeneration (AMD) is the leading cause of blindness in individuals over 65 in developed countries [17]. As the name implies, AMD primarily affects the central retina or macula. It is characterized by the development of drusen, extracellular lipoprotein deposits under the retinal pigment epithelium (RPE), in the early stages of the disease, followed by the loss of photoreceptors and RPE. Choroidal neovascularization (CNV) develops during later stages of the disease (wet AMD) [17]. Choroidal neovascularization (CNV) is the growth of new blood vessels that originate from the choroid through a break in Bruch’s membrane into the subretinal space or sub-retinal pigment epithelium. Dry or atrophic AMD is the most common form of AMD and there are no treatments for this form of AMD.

Aging is a recognized factor that is associated with mitochondrial dysfunction in the outer retina. Other important risk factors for the development of AMD are genetics and environmental stressors, including smoking and exposure to ultraviolet and blue light [18,19,20,21,22,23,24]. All of these factors are likely to result in mtDNA damage in the RPE.

Genetic risk factors also contribute to the development of AMD. Genetic polymorphisms of complement factor H (CFH) contribute to AMD pathology. An important relationship between mitochondrial dysfunction and CFH has been recently established [24,25]. These studies have demonstrated mitochondrial abnormalities in Complement factor H null (Cfh^−/−^) mice that are characterized by disrupted mitochondrial morphology and decreased ATP production [24,25]. These findings suggest that CFH plays an important role in retinal oxidative metabolism. Genome wide associated studies have implicated several genes in the cholesterol pathway, including apolipoprotein E (APOE), in the development of AMD [17]. Recent studies by Lee et al. (2016) have shown that mutations in the transmembrane 135 (Tmem 135) gene that codes for a mitochondrial protein located in RPE and photoreceptors results in accelerated retinal aging in a mouse model of AMD [26].

Although in vitro studies have unequivocally demonstrated that either UV or blue light damage RPE cells [22], epidemiological evidence of the association between exposure to sunlight and AMD is mixed [23]. The ability of the ocular media (cornea, lens, and vitreous humor) to block UV and blue light and the protection of the retina by antioxidant systems, including superoxide dismutase and glutathione peroxidase and macular pigments, including melanin and flavoproteins likely explain these differences [17].

Recent studies suggest that a bioenergetic crisis in the RPE contributes to the pathology of AMD, as illustrated in Figure 3 [20]. Analysis of retinal tissue from human donors with AMD have documented a reduction in the mitochondrial number, a disruption in mitochondrial structure and an increase in mtDNA damage in the RPE that correlates with the severity of disease [15,16,17,18]. Recent studies in the primary cultures of RPE derived from human donors with and without AMD have shown decreased rates of ATP synthesis from oxidative phosphorylation and glycolysis in AMD donors consistent with a bioenergetic crisis. Fisher and Ferrington (2018) have suggested that mitochondrial damage is limited to the RPE, because these cells rely nearly exclusively on mitochondrial oxidative phosphorylation, whereas photoreceptors utilize aerobic glycolysis [4,5,6,7]. In addition to disruptions in mitochondrial bioenergetics, mitochondrial dysfunction has been shown to disrupt calcium homeostasis and mitochondrial nuclear signaling. Several studies have suggested that therapeutic approaches that target RPE mitochondria may provide an effective early treatment strategy for AMD [18,19,20,21,22]. 

### 2.3. Diabetic Retinopathy

Diabetic Retinopathy (DR) is the most common complication of diabetes mellitus. DR is a leading cause of blindness in many developed countries [27]. Chronic hyperglycemia coupled with other risk factors, including hypertension and dyslipidemia, are postulated to trigger a cascade of biochemical and pathophysiological changes that lead to microvascular damage and retinal dysfunction. Mitochondrial dysfunction is a key component of this cascade.

The pathogenesis of diabetic retinopathy is complex, involving the disruption of multiple intracellular pathways [27]. Early pathologic changes that were observed in diabetic retinopathy include mitochondrial dysfunction, oxidative stress, and inflammation [27,28,29,30]. Oxidative stress results from the increased production of reactive oxygen species (ROS), including superoxide and hydrogen peroxide. ROS oxidize intracellular proteins, lipids, and nucleic acids, disrupting normal signaling and culminating in disease. Under physiological conditions, small amounts of superoxide leak from the electron transport chain and they are converted to hydrogen peroxide by mitochondrial superoxide dismutase. Hydrogen peroxide diffuses out of the mitochondria into the cytosol and it serves an important signaling molecule. Intracellular ROS concentrations are regulated by a complex array of antioxidant systems to maintain low intracellular ROS concentrations [29]. Under hyperglycemic conditions, the excess production of ROS overwhelms the antioxidant systems, resulting in oxidative damage [11,28,29].

Hyperglycemia is known to disrupt several metabolic pathways that are involved in the pathogenesis of diabetic retinopathy. These pathways include protein kinase C (PKC) activation, accumulation of advanced glycation products (AGEs), the polyol pathway, and activation of the hexosamine pathway [27,28]. Experimental models have shown that, in the etiology of this progressive disease, the activation of NADPH oxidase (*Nox*) increases cytosolic ROS and this sustained accumulation of ROS damages mitochondria, which further increases oxidative stress [30]. Superoxide radicals (O_2_^−^) that are produced by the mitochondrial electron transport chain activate these major pathways, and activation, in turn, can damage the mitochondria propagating the vicious cycle of ROS.

A major source of the excess ROS production under hyperglycemic conditions is the mitochondrial electron transport chain (ETC) [28,29,30]. Under normal physiological conditions, the electrons are donated to complex I or complex II of the ETC and are then passed on to coenzyme Q, complex III, cytochrome C, complex IV, and ultimately to molecular oxygen, the final electron acceptor. The transfer of electrons generates a proton gradient across the mitochondrial intermembrane, which drives the synthesis of ATP by ATP synthase. During this process, small amounts of superoxide are produced, but the cell via antioxidant defense mechanisms easily clears them [20,21,22,23,24,25,26,27,28,29,30].

Complex I of the ETC is generally regarded as the primary source of superoxide production. However, the 2-oxoglutarate dehydrogenase (OGDH), branched-chain 2-oxoacid dehydrogenase (BCKDH), and pyruvate dehydrogenase (PDH) complexes are also capable of considerable superoxide/H_2_O_2_ production. Moreover, mitochondria can generate ROS, superoxide, or hydrogen peroxide, from at least ten distinct sites on the ETC or associated pathways [31,32,33,34].

The precise cause of mitochondrial dysfunction in a diabetic state is unclear. Some studies suggest that the increased glucose concentration overwhelms the electron transport chain by increasing the rate of oxidative phosphorylation [16,17,18,19], thus generating ROS. Other studies suggest that mitochondrial DNA is damaged, which results in dysfunctional proteins that are required for ETC complexes, also causing excess ROS and mitochondrial dysfunction [27,28].

Mitochondrial function, structure, and mtDNA are damaged in the retina in diabetic retinopathy [29]. Mitochondrial biogenesis and mtDNA repair mechanisms are compromised [35,36]. The ETC becomes dysfunctional and the import of nuclear DNA encoded proteins is disrupted. These diabetes-induced abnormalities in mitochondria persist after the removal of the hyperglycemic insult and they are implicated in the metabolic memory phenomenon associated with the continued progression of diabetic retinopathy [36,37]. Hyperglycemic insult also results in epigenetic modifications [38]. These epigenetic changes in histones and DNA methylation can be passed to the next generation. Epigenetic modifications also contribute to mitochondrial damage and disrupt mitochondrial homeostasis [38]. Therapeutic strategies directed at mitochondrial homeostasis and epigenetic modifications have the potential to halt or slow the progression of diabetic retinopathy.

### 2.4. Glaucoma

Glaucoma is a leading cause of blindness, which is characterized by the accelerated death of retinal ganglion cells (RGC), leading to vision loss [39,40,41,42]. Age and elevated intraocular pressure (IOP) are two major risk factors in developing glaucoma. Therapies reducing IOP have been shown to slow the progression of glaucoma in many but not all patients. Aging may increase the vulnerability of the optic nerve to various stressors, ultimately resulting in RGC death and optic nerve degeneration [39,40,41].

A growing body of evidence supports a key role for mitochondrial dysfunction in the pathogenesis of glaucoma [39,40,41,42,43,44,45,46,47,48,49,50,51]. Mitochondrial dysfunction has been demonstrated in RGC loss in animal and cultured cell experimental models of glaucoma [47,51]. Mutations in mitochondrial and nuclear genes encoding mitochondrial proteins are known to cause primary optic neuropathies, including Leber's Hereditary Optic Neuropathy (LHON) and Autosomal Dominant Optic Atrophy (AODA) [51]. Recent studies indicate that the bioenergetic consequences of mtDNA and nuclear DNA mutations contribute to the development of Primary Open Angle Glaucoma (POAG) and Normal Tension Glaucoma (NTG) [40,41,42,43].

In addition to genetic factors, retinal ganglion cells are profoundly susceptible to mitochondrial dysfunction [42,51]. A combination of the acute energy demand of RGCs and their unique morphology appears to underlie the susceptibility of these neurons to mitochondrial dysfunction. The RGC cell body is located in the ganglion cell layer of the retina. RGCs possess elaborate dendritic arbors that project into the inner plexiform layer to synapse with other retinal neurons. These dendritic arbors are packed with mitochondria that are needed to supply the ATP required to maintain these connections. The long axons of the RGCs form the optic nerve and extend into the brain. The RGC axons make a 90-degree turn at the optic nerve head and pass through the lamina cribrosa, a series of perforated collagen plates, before forming the optic nerve. After passing through the lamina cribrosa, the optic nerve becomes laminated and continues to the visual cortex. The optic nerve has one of the highest oxygen consumption rates and energy demands of any tissue in the body [51]. The mitochondrial density is greater in the unmyelinated region of the optic nerve than in the myelinated region. The unmyelinated prelaminar and laminar regions of the optic nerve require more energy than the myelinated segment of the optic nerve due to the absence of saltatory conduction. This region of the optic nerve has a high density of voltage gated sodium channels and it requires more energy to restore ion gradients.

The combination of genetic susceptibility due to inherited or acquired mutations, energy demand, and unique morphology of retinal ganglion cells in an aging population all contribute to the pathogenesis and incidence of glaucoma (Figure 4).

## 3. Conclusions

Experimental evidence has shown a link between mitochondrial dysfunction and the loss of vision. Most mitochondrial diseases exhibit some form of visual impairment. Many retinal and optic nerve diseases, including AMD, diabetic retinopathy, and glaucoma are also characterized by mitochondrial dysfunction. The development of therapeutic approaches that target retinal mitochondrial dysfunction has the potential to profoundly impact the treatment of retinal and optic nerve disease.

## Figures and Tables

**Figure 1 biology-08-00031-f001:**
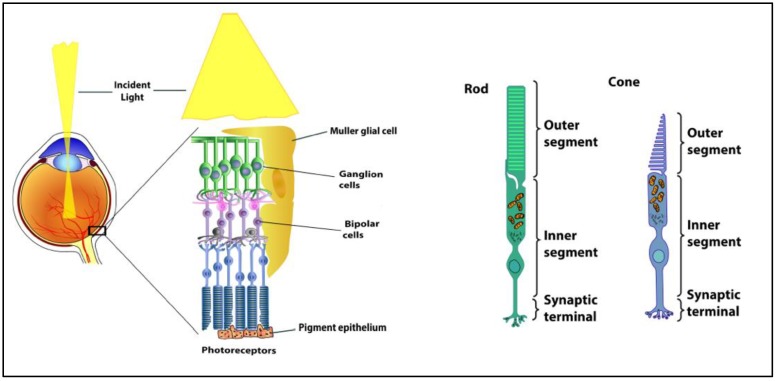
Anatomy of the retina and the structure of rod and cone photoreceptors.

**Figure 2 biology-08-00031-f002:**
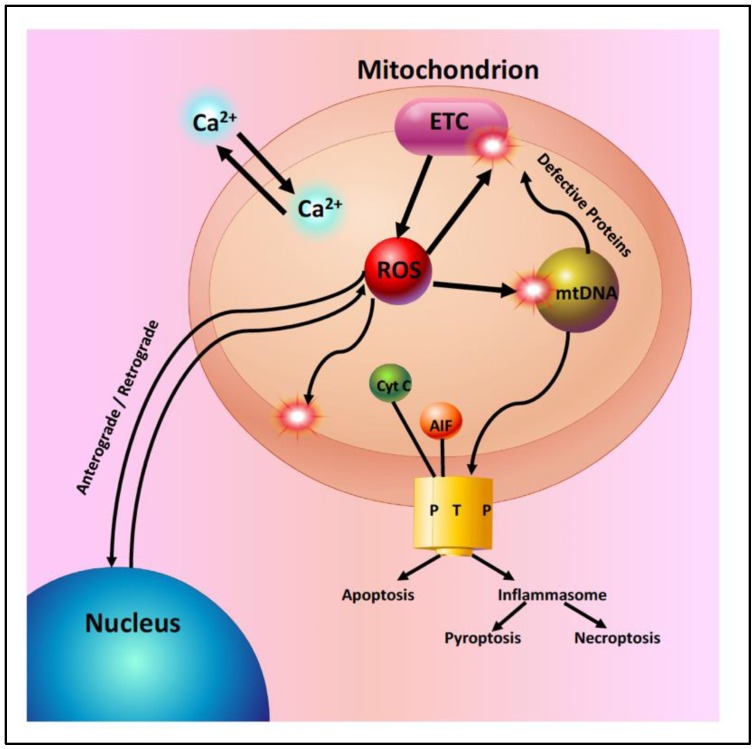
Mitochondria in the aging retina. Mitochondria are essential for many cellular functions including: (1) the synthesis of ATP by oxidative phosphorylation, (2) the regulation of intracellular calcium homeostasis, (3) anterograde and retrograde signaling between the nucleus and mitochondria, (4) the generation of reactive oxygen species (ROS) from the electron transport chain (ETC). ROS act as signaling molecules in low concentrations or as toxic molecules in higher concentrations. ROS oxidize mitochondrial lipids, proteins and DNA and (5) the regulation of apoptosis. Excess ROS or intramitochondrial calcium can lead to the activation of cell death pathways by opening the mitochondrial permeability transition pore (PTP).

**Figure 3 biology-08-00031-f003:**
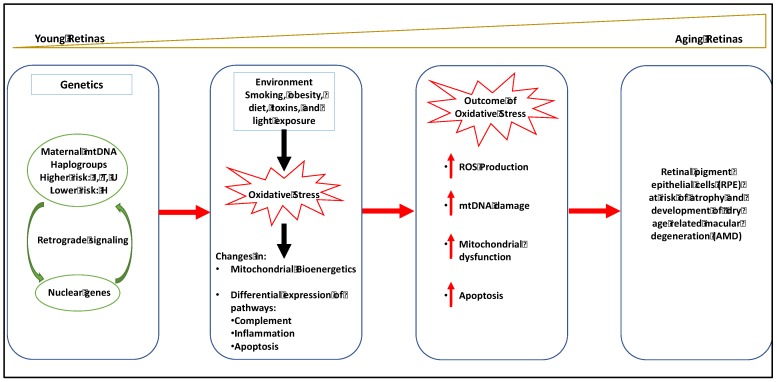
Mitochondrial involvement in dry age-related macular degeneration (AMD).

**Figure 4 biology-08-00031-f004:**
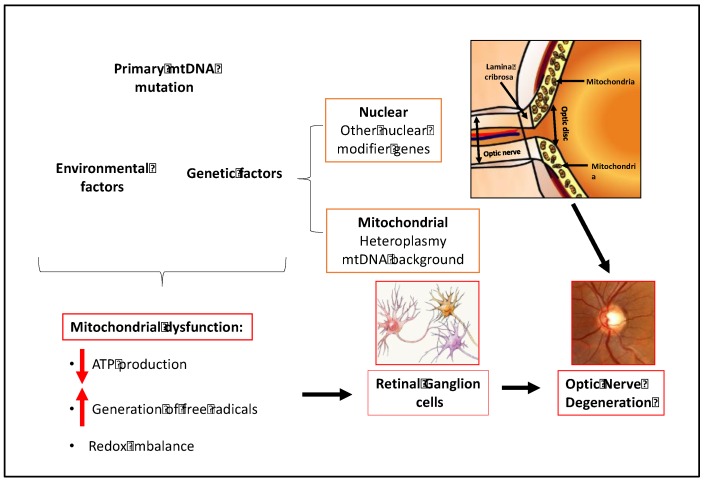
Mitochondrial dysfunction in glaucoma.

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
