# Peer review of "Mitochondrial Dysfunction in the Aging Retina"

_biology, 2019, doi:10.3390/biology8020031_

Reviewer 1 Report

This is a well written review of the role of mitochondria in several common diseases affecting vision. The author, Dr. Ells, covers the main topics and diseases of interest.  Nevertheless, the review relies too heavily on other earlier reviews rather than on primary research literature.  This severely limits the utility of this paper, since old reviews will have cited even older research papers.  The author should cite recent research papers to substantiate the points she is making.

Other points:

P. 1 line 38, Mention that a large fraction of retinal ATP is derived from aerobic glycolysis.

P.2 The author should be careful about using the acronym “ROS” to stand for reactive oxygen species, since in the field of retinal biology the same term stands for "rod outer segments". At least the term should be defined the first time it is used.

P.2 line 61, Kennedy and colleagues have shown that mtDNA mutations that accumulate with age are probably the result of replication error, not oxidative stress. PMID: 24086148.

P. 3 line 80-81, define “drusen” and “choroidal neovascularization”.

P.3 line 86: Most UV light is filtered out by the lens and cornea.  Reference 8 presents no evidence on blue light. It is simply a review on mitochondrial dysfunction in retinal disease.  The author should cite primary research, not another review. For example King and colleagues (PMID: 15191057)  showed that blue light injured RPE cells and that this effect could be blocked by mitochondrial antioxidants.

P.3  line 89: Because of linkage disequilibrium between ARMS2 and HTRA1, it is impossible to state with confidence that a single mutation in the ARMS2 locus confers risk of developing AMD. The localization of ARMS2 protein is controversial, so to call it unambiguously mitochondrial may mislead the readers.

P. 5 Line 139: Flavin dependent enzymes in the mitochondrial matrix also lead to the production of superoxide and H2O2. See: PMID:  24515115.

P.5 line 146: The author may want to discuss the metabolic memory hypothesis as it relates to diabetic retinopathy.

P. line 147: The references concerning the role of mitochondria in primary open angle glaucoma are not up to date.  Recent primary references should be cited.

Author Response

The author would like to thank both reviewers for detailed and exceptional reviews.  The manuscript is much improved by your efforts.

RESPONSE TO REVIEWER 1

POINT 1a. This is a well written review of the role of mitochondria in several common diseases affecting vision. The author, Dr. Eells, covers the main topics and diseases of interest.  Nevertheless, the review relies too heavily on other earlier reviews rather than on primary research literature.  This severely limits the utility of this paper, since old reviews will have cited even older research papers.  The author should cite recent research papers to substantiate the points she is making.

Response 1a:    Thank you.   More recent primary research publications have been incorporated.

Other points:

P. 1 line 38, Mention that a large fraction of retinal ATP is derived from aerobic glycolysis.

Response 2:  This section has been modified.

The retina is the highest oxygen-consuming organ in the human body [2-6]. Inner retinal neurons have the highest metabolic rate of all central nervous tissue, and the oxygen consumption rate of the photoreceptors is, several times higher again. The inner segments of the photoreceptor cells are rich in mitochondria providing the ATP required by the ionic pumps which drive the ‘dark current.  In addition, photoreceptors also metabolize glucose through aerobic glycolysis [2-6].

The key role of aerobic glycolysis in the vertebrate retina has been long recognized [2-6] (Warburg 1927) and well established [2,5,6].  The majority of aerobic glycolysis occurs in the photoreceptors. Although aerobic glycolysis dominates energy production in photoreceptors, the retina requires both glycolysis and oxidative phosphorylation to initiate vision [4].  This has been demonstrated in studies showing that the responses of neurons downstream of photoreceptors are abolished either by inhibiting glycolysis or by removing O2 [2].  Recent studies have proposed a model in which glucose from the choroidal blood passes through the retinal pigment epithelium to the retina where photoreceptors convert it to lactate [6].  Photoreceptors then export the lactate as fuel for the retinal pigment epithelium and for neighboring Müller glial cells. Aerobic glycolysis not only enables enhanced anabolic metabolism in photoreceptors, but also maximizes their function and adaptability to nutrient stress conditions.

Photoreceptors consume more oxygen per gram of tissue weight than any cell in the body [2,5].   This intense degree of oxidative phosphorylation in their inner segments, coupled with high concentrations of polyunsaturated fatty acids in their outer segments renders the retina susceptible to oxidative stress and lipid peroxidation [2-4].   The retina is exposed to visible light and contains abundant photosensitizers [1]. Normally, oxidative damage is minimized by endogenous antioxidants and repair systems.  However, with aging and retinal disease there is an increase in mitochondrial dysfunction and oxidative damage and corresponding decrements in antioxidants and repair systems, resulting in retinal dysfunction and retinal cell loss, leading to visual impairment [11-15]

P.2 The author should be careful about using the acronym “ROS” to stand for reactive oxygen species, since in the field of retinal biology the same term stands for "rod outer segments". At least the term should be defined the first time it is used.

Response 3:   ROS has been defined as “reactive oxygen species”

P.2 line 61, Kennedy and colleagues have shown that mtDNA mutations that accumulate with age are probably the result of replication error, not oxidative stress. PMID: 24086148.

Response 4:  The text has been revised to read:

Mitochondria in the aging retina show a decline in the function of the electron transport chain and a significant increase in the generation of reactive oxygen species (ROS).(Figure 2) This increased ROS production leads to increased oxidative damage to mitochondrial DNA, lipids and proteins [4-9].  With aging there is an accumulation of mtDNA mutations and a disruption of mitochondrial structure.  Mitochondrial genomic instability has been postulated to play an important role in age-related retinal pathophysiology [9] However, recent studies have shown that the majority of mtDNA mutations in aged cells appear to be caused by replication errors early in life, rather than by oxidative damage [8]. Mitochondrial DNA repair pathways are limited and begin to fail with aging leading to additional mtDNA damage and contributing to the development of retinal degenerative diseases [11].

P. 3 line 80-81, define “drusen” and “choroidal neovascularization”.

Response 5:   Drusen have been defined as  extracellular lipoprotein deposits under the RPE

Choroidal Neovascularization (CNV) has been defined as is the growth of new blood vessels originating from the choroid through a break in Bruch’s membrane into the subretinal space or sub-retinal pigment epithelium.

P.3 line 86: Most UV light is filtered out by the lens and cornea.  Reference 8 presents no evidence on blue light. It is simply a review on mitochondrial dysfunction in retinal disease.  The author should cite primary research, not another review. For example King and colleagues (PMID: 15191057) showed that blue light injured RPE cells and that this effect could be blocked by mitochondrial antioxidants.

Response 6:  Corrected the reference and added appropriate references from the primary literature. Now reads as follows:

Although in vitro studies have unequivocally demonstrated that either UV or blue light damage RPE cells [21], epidemiological evidence of the association between exposure to sunlight and AMD is mixed [23]. These differences are likely explained by the ability of the ocular media (cornea, lens and vitreous humor) to block UV and blue light and the protection of the retina by antioxidant systems including superoxide dismutase and glutathione peroxidase and macular pigments including melanin and flavoproteins [22].  

P.3 line 89: Because of linkage disequilibrium between ARMS2 and HTRA1, it is impossible to state with confidence that a single mutation in the ARMS2 locus confers risk of developing AMD. The localization of ARMS2 protein is controversial, so to call it unambiguously mitochondrial may mislead the readers.

Response 7:  Have removed this discussion from the text

P. 5 Line 139: Flavin dependent enzymes in the mitochondrial matrix also lead to the production of superoxide and H2O2. See: PMID:  24515115.

Response 8:   Incorporated this reference and related primary literature references and added the following text to the manuscript:

Complex I of the ETC is generally regarded as the primary source of superoxide production. However, the 2-oxoglutarate dehydrogenase (OGDH), branched-chain 2-oxoacid dehydrogenase (BCKDH), and pyruvate dehydrogenase (PDH) complexes are also capable of considerable superoxide/H2O2 production. Moreover, mitochondria can generate ROS, superoxide or hydrogen peroxide, from at least ten distinct sites on the ETC or associated pathways [31-34]

P.5 line 146: The author may want to discuss the metabolic memory hypothesis as it relates to diabetic retinopathy.

Response 9:  Added discussion of metabolic memory as relates to diabetic retinopathy.

Mitochondrial function, structure and mtDNA are damaged in the retina in diabetic retinopathy [29]. Mitochondrial biogenesis and mtDNA repair mechanisms are compromised [35,36]. The ETC becomes dysfunctional and the import of nuclear DNA encoded proteins is disrupted. These diabetes-induced abnormalities in mitochondria persist after the removal of the hyperglycemic insult and are implicated in the metabolic memory phenomenon associated with the continued progression of diabetic retinopathy [36-37]. Hyperglycemic insult also results in epigenetic modifications [38].  These epigenetic changes in histones and DNA methylation can be passed to the next generation. Epigenetic modifications also contribute to mitochondrial damage and disrupt mitochondrial homeostasis [38]. Therapeutic strategies directed a mitochondrial homeostasis and epigenetic modifications have the potential to halt or slow the progression of diabetic retinopathy

P. line 147: The references concerning the role of mitochondria in primary open angle glaucoma are not up to date.  Recent primary references should be cited.

Response 10:  Thank you.  I have added more primary literature references to support the text.

Reviewer 2 Report

This review is suitable for the journal. I have only minor comments.

1] I do not think it is correct t to state “Modulation of the dark current is the beginning of vision”

2] The author needs to discuss the role of glycolysis more extensively.  There is evidence that glycolysis increases with age as mitochndrial function declines. Also, glycolysis increases the pace of ageing via AGEs production.

3] Is mitochndrial genomic stability as big an issue as we think? There is evidence that mutations are simply an amplification of errors present early in life – Reviewed in Lopez-Otin et al Cell 2013

4] There is now a direct association between polymorphisms of complement  and mitochndrial decline in models of AMD;  Sci Rep. 2019 Jan 31;9(1):1082. doi: 10.1038/s41598-018-37673-6.

Author Response

RESPONSE TO REVIEWER #2

Top of Fo

This review is suitable for the journal. I have only minor comments.

1] I do not think it is correct t to state “Modulation of the dark current is the beginning of vision”

Response 1:  In agreement with the reviewer, this sentence has been removed.

2] The author needs to discuss the role of glycolysis more extensively.  There is evidence that glycolysis increases with age as mitochondrial function declines. Also, glycolysis increases the pace of ageing via AGEs production.

Response 2: The text has been revised as follows:

The key role of aerobic glycolysis in the vertebrate retina has been long recognized and well established [2,5,6].  The majority of aerobic glycolysis occurs in the photoreceptors. Although aerobic glycolysis dominates energy production in photoreceptors, the retina requires both glycolysis and oxidative phosphorylation to initiate vision [4].  This has been demonstrated in studies showing that the responses of neurons downstream of photoreceptors are abolished either by inhibiting glycolysis or by removing O2 [2].  Recent studies have proposed a model in which glucose from the choroidal blood passes through the retinal pigment epithelium to the retina where photoreceptors convert it to lactate [6].  Photoreceptors then export the lactate as fuel for the retinal pigment epithelium and for neighboring Müller glial cells. Aerobic glycolysis not only enables enhanced anabolic metabolism in photoreceptors, but also maximizes their function and adaptability to nutrient stress conditions.

3] Is mitochondrial genomic stability as big an issue as we think? There is evidence that mutations are simply an amplification of errors present early in life – Reviewed in Lopez-Otin et al Cell 2013

Response 3:  The following text has been added.

Mitochondria in the aging retina show a decline in the function of the electron transport chain and a significant increase in the generation of reactive oxygen species (ROS). (Figure 2) This increased ROS production leads to increased oxidative damage to mitochondrial DNA, lipids and proteins [4-9].  With aging there is an accumulation of mtDNA mutations and a disruption of mitochondrial structure.  Mitochondrial genomic instability has been postulated to play an important role in age-related retinal pathophysiology [9] However, recent studies have shown that the majority of mtDNA mutations in aged cells appear to be caused by replication errors early in life, rather than by oxidative damage [8]. Mitochondrial DNA repair pathways are limited and begin to fail with aging leading to additional mtDNA damage and contributing to the development of retinal degenerative diseases [11].

4] There is now a direct association between polymorphisms of complement and mitochondrial decline in models of AMD;  Sci Rep. 2019 Jan 31;9(1):1082. doi: 10.1038/s41598-018-37673-6.

Response 4:  Now Discussed as follows:

Genetic risk factors also contribute to the development of AMD. Genetic polymorphisms of complement factor H (CFH) contribute to AMD pathology. An important relationship between mitochondrial dysfunction and CFH has been recently established (REFS) These studies have demonstrated mitochondrial abnormalities in Complement factor H null (Cfh/) mice characterized by disrupted mitochondrial morphology and decreased ATP production [24,25].  These findings suggest that CFH plays an important player in retinal oxidative metabolism.  Genome wide associated studies have implicated several genes in the cholesterol pathway including apolipoprotein E (APOE) in the development of AMD. Recent studies by Lee et al (2016) have shown that mutations in the transmembrane 135 (Tmem 135) gene that codes for a mitochondrial protein located in RPE and photoreceptors results in accelerated retinal aging in a mouse model of AMD [26].

Biology EISSN 2079-7737 Published by MDPI AG, Basel, Switzerland RSS E-Mail Table of Contents Alert
Back to Top